# Immunometabolism of Immune Cells in Mucosal Environment Drives Effector Responses against *Mycobacterium tuberculosis*

**DOI:** 10.3390/ijms23158531

**Published:** 2022-08-01

**Authors:** Mohd Hatimi Tukiman, Mohd Nor Norazmi

**Affiliations:** School of Health Sciences, Universiti Sains Malaysia (USM), Kubang Kerian, Kota Bharu 16150, Kelantan, Malaysia; hatimi.tukiman@gmail.com

**Keywords:** tuberculosis, immunometabolism, innate and adaptive immune cells, glycolysis, oxidative phosphorylation, gut-lung axis

## Abstract

Tuberculosis remains a major threat to global public health, with more than 1.5 million deaths recorded in 2020. Improved interventions against tuberculosis are urgently needed, but there are still gaps in our knowledge of the host-pathogen interaction that need to be filled, especially at the site of infection. With a long history of infection in humans, *Mycobacterium tuberculosis* (Mtb) has evolved to be able to exploit the microenvironment of the infection site to survive and grow. The immune cells are not only reliant on immune signalling to mount an effective response to Mtb invasion but can also be orchestrated by their metabolic state. Cellular metabolism was often overlooked in the past but growing evidence of its importance in the functions of immune cells suggests that it can no longer be ignored. This review aims to gain a better understanding of mucosal immunometabolism of resident effector cells, such as alveolar macrophages and mucosal-associated invariant T cells (MAIT cells), in response to Mtb infection and how Mtb manipulates them for its survival and growth, which could address our knowledge gaps while opening up new questions, and potentially be applied for future vaccination and therapeutic strategies.

## 1. Introduction

Mortality associated with tuberculosis (TB), caused by *Mycobacterium tuberculosis* (Mtb), recorded a slight increase from a total of 1.4 million TB deaths in 2019 to 1.5 million in 2020 [1]. The rise in mortality may be due to the impact of the global COVID-19 pandemic that had reduced the access to TB diagnosis and treatment [1,2]. This has further dented our progress in the WHO End TB Strategy towards the elimination of the global TB epidemic, which had already been shown to be lagging even before the COVID-19 pandemic [1]. These impacts are predicted to be worse in the coming years as COVID19 co-infection is highly likely, with the estimation that a quarter of the global population is latently infected with Mtb [3]. COVID-19 has been shown to impair the function of immune cells [4,5], which could have a similar effect that favours TB disease progression as seen in other conditions such as human immunodeficiency virus (HIV) co-infection, initiation of anti-tumour necrosis factor therapy, and malnutrition [6,7]. Thus, improved efforts are urgently needed to significantly reduce the global TB burden.

Even with great advances in the field of immunology in the past 30 years, there are still many gaps in our understanding of the entire mechanism of Mtb infection, which contributes to the current inefficiencies in controlling TB. In recent years immunologists have started to integrate a metabolomic approach into their research, creating an extended field termed immunometabolism [8]. Research in TB immunometabolism is gathering pace as an increasing number of studies in this field are published each year. In this review, we will discuss both early and chronic phases of the immune response to Mtb at the site of infection, and how metabolism influences those responses. With emerging evidence of a possible association of the gut-lung axis with pulmonary TB, factors from another site of the mucosal network will also be considered. Thus, this review will also provide a glimpse into the potential role of the gut-lung axis in exploring the mechanism of host immunity to TB. Understanding the cellular interplay between immune responses and metabolism during Mtb infection will generate ideas for improved TB vaccination strategies and new targets for therapy.

## 2. Immunometabolism during Early Phase of Mtb Infection

Innate immune cells that will be discussed in this review include alveolar epithelial cells, macrophages, dendritic cells, neutrophils and mucosal-associated invariant T (MAIT) cells. An overview of the immunometabolism of these cells in response to Mtb infection is illustrated in Figure 1.

### 2.1. Alveolar Epithelial Cells

Inhaled Mtb must first breach the alveolar barrier to infect its host. The alveolar lumen is lined mostly (90–95%) by type 1 alveolar epithelial cells (AECs), which primarily facilitate gas exchange in the lungs, and the remaining 5–10% is occupied by type 2 AECs that are responsible for secreting surfactants and restoring damaged type 1 AECs [9,10]. Recent findings suggest that Mtb can invade and replicate in type 2 AECs, which are the favourable hiding place for Mtb from immune cells, as type 2 AECs are non-professional phagocytes [11]. This is reflected by the transcriptome of Mtb in type 2 AECs, which showed upregulation of genes for replication, cell wall synthesis, aerobic respiration, and virulence [12]. In addition, Mtb genes involved in alternative electron transfer and non-aerobic respiration, and genes encoding Universal Stress Protein and other hypoxia-induced genes are downregulated, indicating a favourable condition for intracellular Mtb growth in AECs [12].

The direct impact of Mtb invasion on AEC metabolism is relatively unknown. Type 2 AECs are known to be the main producer of pulmonary surfactants and they need to produce lipids even under metabolically unfavourable conditions [13]. This could be another reason why type 2 AEC is a good hiding place for Mtb. Altered surfactant functions could lead to rapid Mtb growth in both AECs and macrophages, as demonstrated recently [14]. Compromised lung health, as may be the case in smoking, may facilitate direct infection of these pulmonary surfactant producers by Mtb, although this remains to be proven.

### 2.2. Macrophages

Macrophages reside in all tissues in the human body, operating as the primary phagocytes [15]. They are termed according to their location, such as alveolar macrophages (AMs) in the alveoli compartment of the lungs and interstitial macrophages (IMs) in the periphery [16,17].

AMs are the primary immune cells facing Mtb infection at the entry point. The nature of the alveoli microenvironment, which is constantly exposed to commensals and environmental components in the inhaled air, leads to a more immunotolerant M2 phenotype of AMs at the basal state. This immunotolerant state is driven by transforming growth factor β (TGF-β) secreted by AECs and AMs that can also induce regulatory signalling by naïve T cells [18,19], which favours Mtb entry and survival. As long-lived and self-renewal cells, AMs are reliant on oxidative phosphorylation (OXPHOS) to support the demands of their phagocytic function. OXPHOS is mainly fuelled by fatty acid oxidation (FAO), as lipids are abundantly available in the lung compared to glucose. This lipid-rich environment is permissive to Mtb infection as the mycobacteria also utilise lipid as their main carbon source [20].

Xenophagy is host-directed autophagy to control intracellular pathogens including Mtb, which is mediated by the binding of host ubiquitin to Mtb surface proteins and subsequent recognition by autophagy receptors [21]. Inhibition of FAO can also promote xenophagy, as demonstrated by Chandra et al., wherein Mtb is unable to grow in macrophages when host FAO is blocked, either chemically by trimetazidine (a compound in clinical use) or genetically by deletion of the mitochondrial fatty acid transporter carnitine palmitoyltransferase 2 (CPT2) [22]. FAO blockage leads to the accumulation of mitochondrial reactive oxygen species, which promotes NADPH oxidase recruitment to the phagosomal membrane, resulting in the induction of xenophagy [22].

In response to Mtb infection, AMs shift their metabolism to mount the antimicrobial and pro-inflammatory responses (M1 phenotype). IMs, which predominantly exhibit the M1 phenotype, are also recruited to the site of infection. M1 macrophages showed similar metabolic characteristics seen in cancer cells, termed the Warburg effect, such as enhanced aerobic glycolysis with the formation of lactate and decreased OXPHOS [23]. Aerobic glycolysis accelerates ATP production, though less efficiently, and provides biosynthetic capacity for activation of immune cells [23]. Gleeson et al. demonstrated that inhibition of AM metabolic shift from OXPHOS to aerobic glycolysis leads to reduction of pro-inflammatory IL-1β and transcription of prostaglandin-endoperoxide synthase 2 (PTGS2), and increased levels of anti-inflammatory IL-10, resulting in increased Mtb survival. They further showed that control of intracellular Mtb replication through induction of AM aerobic glycolysis is dependent on IL-1β signalling [24].

Mtb has been shown to evade host immunometabolic responses by limiting metabolic reprogramming of macrophages through induction of anti-inflammatory microRNA-21 (miR-21), which suppresses glycolysis and limits pro-inflammatory mediators such as IL-1β by targeting phosphofructokinase muscle (PFK-M) isoform [25]. miR-21 in turn is targeted by interferon gamma (IFN-γ), produced by activated immune cells, to induce macrophage glycolysis to support pro-inflammatory activities [25]. In another study, Rahman et al. showed that Mtb infection in mouse peritoneal macrophages induced host hydrogen sulphide (H_2_S) production, which leads to suppression of glycolysis. Cystathionine γ-lyase (CSE) is one of the enzymes responsible for H_2_S synthesis, and Mtb infection in CSE-deletion peritoneal macrophages resulted in a two- to three-fold increase in glycolytic intermediates with a lower bacterial burden as compared to wild-type macrophages [26].

### 2.3. Dendritic Cells

Dendritic cells (DCs) are the vital link between the innate and adaptive immune systems of the host in response to TB. DCs take up live Mtb or the bacterial antigen at the site of infection and travel to lymph nodes to activate adaptive immune cells. In addition to their task as antigen-presenting cells, activated DCs also produce a significant amount of pro-inflammatory cytokines such as IL-6, TNF-α, and IL-1β [27].

Immature DCs have elevated OXPHOS, fatty acid metabolism, and mitochondrial biogenesis [27]. Mtb infection activates the Tank binding kinase (TBK) 1, protein kinase Akt, and hexokinase-2 signalling pathways that lead to DC maturation [28]. Subsequently, molecules involved in mammalian target of rapamycin (mTOR) and hypoxia-inducible factor 1α (HIF-1α) are upregulated, which elevate glycolysis, fatty acid synthesis, and the pentose phosphate pathway to fuel anti-microbial responses against Mtb infection [27].

### 2.4. Neutrophils

Neutrophils are the most abundant type of white blood cells and among the earliest cells to arrive at the site of Mtb infection, but their specific role is less known to date [27]. Neutrophils are efficient phagocytes and recruit other inflammatory monocytes from circulation through the secretion of cytokines and chemokines after phagocytosis of mycobacteria [29,30]. Neutrophils can also contribute to the exacerbation of inflammation and disease progression, demonstrated by increased neutrophil matrix metalloproteases (MMP)-8 and MMP-9, and neutrophil elastase secretion [31]. These events are mediated by a hypoxic environment induced by Mtb infection and dependent on the neutrophil HIF-1α pathway [31]. The role of these proteases in protection against TB is unknown, but elastase has been demonstrated to have anti-mycobacterial activity [32]. The role of neutrophils in response to Mtb infection needs further clarification to modulate their function and reduce disease severity.

### 2.5. MAIT Cells

Mucosal-associated invariant T (MAIT) cells are the most abundant innate-like T cell population in humans, comprising up to ~5% of the total T cell population [33]. MAIT cells are characterised by the expression of the conserved T cell receptor (TCR) α-chain Vα7.2 s(TRAV 1-2) in humans with oligoclonal Vβ chain usage, which responds to microbially derived non-peptide antigens, namely vitamin B metabolite intermediates presented by the MHC-1-related protein MR1 [34]. Activation of the MAIT cell through its TCR leads to either the expression of pro-inflammatory cytokines such as TNF-α, IFN-γ, and IL-17, or the release of cytotoxic and pro-inflammatory perforin and granzyme B [33]. Thus, MAIT cells are capable of mounting an antimicrobial response and killing infected cells upon activation.

In humans, the highest frequency of MAIT cells is found in blood and liver, but they are also present in the mucosal barriers such as the lung and intestine [35]. Upon Mtb infection at the mucosal site, MAIT cells are shown to be enriched, as demonstrated by Wong et al. in the bronchioalveolar lavage samples of active TB patients [36]. In contrast, the number of circulating MAIT cells was found to be generally reduced in active TB patients, with improved functionality but not frequency of MAIT cells after 10 weeks of TB treatment [37]. These findings suggest a role played by MAIT cells in response to Mtb infection that needs to be explored.

The mechanism of MAIT cell activation during Mtb infection is poorly defined. Studies by Vorkas et al. showed that priming MAIT cells with the synthetic MR1 ligand, 5-OP-RU, leads to enhanced MAIT cell activation and expansion [38]. However, these MAIT cells could not stop Mtb infection and growth, suggesting that MAIT cell priming via MR1 alone is not sufficient to control Mtb infection [38]. Thus, MAIT cells may need additional signals for their complete activation to mount an effective immune response, which needs further investigation.

Metabolic properties of MAIT cells have only started to be investigated in recent years, with not much known to date, especially in TB disease. By integrating gene expression and functional data, Zinser et al. showed that MAIT cells are metabolically quiescent in the resting state, like naïve T cells and central memory T cells, and rapidly enhance their glycolytic activity after stimulation [39]. Tissue-resident MAIT cells may rely on OXPHOS as they adapt to low tissue glucose concentrations [40]. In addition to OXPHOS, IL-17-producing bronchioalveolar lavage MAIT cells from children with community-acquired pneumonia have been shown to be enriched for genes encoding glycolysis and lipid efflux [41]. It is tempting to speculate that activated MAIT cells could also play a role in dampening Mtb growth via increasing their glycolytic activities.

## 3. Immunometabolism during Chronic Phase of TB

### 3.1. Granuloma Formation

As the infection progresses, infected immune cells migrate into the lung interstitial tissue, recruiting more innate immune cells and activated adaptive immune cells such as T cells and B cells that migrate to the infection site [20,42,43]. This focal interplay between Mtb and immune cells leads to granuloma formation. Innate immune cells as discussed above, including natural killer (NK) cells [43], are involved in granuloma formation, together with immune cells activated and differentiated in the latter stages of infection (Figure 2). In addition to M1/M2 polarization, activated macrophages could also differentiate into epithelioid cells, foam cells, or they can fuse to form multinucleated giant cells [44]. Foam cells, or foamy macrophages, and T cells are among the most studied types of immune cells regarding immunometabolism, thus only these cells will be elaborated on further in the next section.

### 3.2. Foamy Macrophages

Alteration of the metabolic profile of Mtb-infected macrophages leads to their transformation into foamy macrophages, which contain high levels of lipid droplets [45,46]. As discussed earlier, macrophages switched to an M1 phenotype upon Mtb infection, with enhanced glycolysis that supports pro-inflammatory responses. However, Mtb can manipulate excessive glycolysis using bacterial factors such as ESAT-6 to elevate lipid accumulation into the macrophage to favour mycobacterial growth [47].

The exact mechanism used by Mtb to enhance the macrophages’ uptake of lipid droplets is currently unclear, but a role for host peroxisome proliferator-activated receptor gamma (PPARγ) and testicular receptor 4 (TR4) has been demonstrated [48]. The PPARγ pathway is upregulated by Mtb infection, and both PPARγ and TR4 induce the level of the macrophage’s scavenger receptor CD36, which takes up exogenous lipid [48]. In turn, the host responds to this by altering lipid metabolism to produce lipid in the forms that are protective against Mtb such as host bioactive lipids prostaglandin E2 and leukotriene B4. This metabolic reprogramming is driven by IFN-γ through HIF-1α [49].

### 3.3. T Cells

T cell-mediated immunity is widely regarded as the most essential adaptive immune response against Mtb infection but less is known about how T cells modulate metabolism to adapt to their response to TB. Increased aerobic glycolysis is associated with T cell activation by T cell receptor ligation and binding of co-stimulatory molecules to induce an anabolic program to increase biomass for proliferation [50]. Subsequently, distinct metabolic programs differentiate T cells into lineages that determine their function. For example, Th1, Th2 and Th17 cells rely heavily on glycolysis to support their functions, expressing a high surface level of glucose transporter Glut1 [51]. In contrast, regulatory T (Treg) cells express low levels of Glut1 and increased lipid oxidation through the AMP-activated protein kinase pathway [51].

CD8 T cells also play an important role in response to Mtb infection by direct killing of infected host cells. However, findings from human and animal studies showed the lack of memory CD8 T cells even after successful treatment, indicating that the development of antigen-experienced CD8 T cells is disrupted during Mtb infection [52]. As Mtb infection persists, CD8 T cells develop bioenergetic deficiencies with a significant reduction in mitochondrial function and increased expression of inhibitory receptors such as CTLA-4 [52].

### 3.4. Modulation of Granuloma for Mtb Dormancy

As illustrated in Figure 2, the granuloma infected by Mtb can be divided into two distinct compartments: the central core area within the granuloma is a hypoxic and pro-inflammatory environment with anti-microbial activity and reactive oxygen species to eliminate Mtb, whereas the peripheral area is associated with anti-inflammatory response [53]. HIF-1α is known to directly promote the inflammatory mediator IL-1β, which is involved in activating glycolytic enzymes [54]. Contrasting levels of HIF-1α expression were observed when both areas were compared, high at the core and low at the periphery [55], which indicates that the immune cells rely on glycolysis for their pro-inflammatory activities at the core area and they utilise OXPHOS for their activities in the peripheral area of the granuloma [55]. However, reduced Warburg effect-associated gene expression was observed at the core of the granuloma, which partly explains Mtb survival and persistence [55]. With this observation, it is speculated that Mtb can modulate the host defence mechanism to dampen the Warburg effect, resulting in a less efficient anti-mycobacterial response [23]. In addition, it is known that Mtb can modulate its metabolism to be dormant in the host cell, with a slow to complete shutdown of replication, and in this state, Mtb is more resistant to anti-mycobacterial agents [56]. Active immune control of Mtb, which may be detectable by the tuberculin skin test (TST) and IFN-γ release assay (IGRA), and persistence of Mtb leads to latent TB infection (LTBI) [6].

LTBI poses a great challenge in our efforts to eliminate TB due to the under-detection of LTBI cases and the unwillingness of people with LTBI to adhere to a lengthy treatment. Recent LTBI treatment has improved from a 9-month isoniazid regimen to a shorter and less frequent regimen such as a 3-month, weekly isoniazid regimen in combination with rifapentine [57,58]. However, concerns regarding the side effects of these drug regimens remained to be addressed, especially hepatotoxicity and other adverse events in patients with co-existing medical conditions such as those requiring hemodialysis [59,60]. Thus, new strategies for LTBI treatment and prevention of active TB disease are urgently needed. With great advancements in recent ‘omics’ technologies, our understanding of Mtb dormancy and its subsequent resuscitation has improved significantly [61], which can help us in designing novel TB vaccines and therapies.

## 4. Modulation of Immunometabolism as Host Directed Therapy for TB Patients and Vaccination Strategy

Modulation of immunometabolism to favour antimicrobial activity by immune cells, especially during the early phases of Mtb infection can potentially halt TB progression. The following are a few examples of potential agents to modulate metabolism in favour of immunity against TB.

Some iron chelators have been shown to modulate cellular metabolism through the regulation of HIF-1α [62]. The iron chelator deferoxamine (DFX) can promote the expression of key glycolytic enzymes in Mtb-infected primary human monocyte-derived macrophages and human AMs and enhance innate immune function by inducing IL-1β in human macrophages during early infection with Mtb and upon stimulation with lipopolysaccharide (LPS) [62].

Suberanilohydroxamic acid (SAHA), an approved histone deacetylase inhibitor (HDACi), can enhance the pro-inflammatory function of human macrophages by promoting an early metabolic switch to glycolysis [63]. SAHA-treated Mtb-infected macrophages have been shown to enhance T helper cell responses but have no effects on cytotoxic T cells [63].

Interestingly, metformin-treated type 2 diabetes patients have a lower risk of Mtb infection, progress from infection to TB disease, TB mortality, and TB recurrence [64]. Expansion of a population of memory-like antigen inexperienced CD8+ CXCR3+ T cells was observed after metformin treatment, with increased (i) mitochondrial mass, OXPHOS, and FAO; (ii) survival capacity; and (iii) anti-mycobacterial properties [64]. CD8 T cell dysfunction associated with chronic Mtb infection is also shown to be reversed after treatment with metformin. Metformin treatment also enhances immunogenicity and protective efficacy against Mtb challenge in BCG-vaccinated mice and guinea pigs [64]. These findings support metformin as a candidate for TB-host directed therapy and as a TB vaccine adjunct.

## 5. A Possible Gut-Lung Axis in TB Protection

Looking beyond the pulmonary area as the site of contact with Mtb, there are reports that gut-associated infections and dysbiosis could influence the immune responses in the lungs during Mtb infection. Although there are currently no reports on immunometabolism, specifically on pathogenesis and dysbiosis in relation to TB, the possibility of gut immunity affecting Mtb infection in the lungs suggests that this field should be explored. Here we will discuss helminth co-infection and dysbiosis of the gut microbiota as evidence of a possible role of the gut-lung axis in defence against TB.

The global burden of parasitic helminth infection is thought to exceed Mtb infection, with an estimated two billion humans being infected in areas that largely overlap with TB endemic countries [65]. Helminth co-infection could enhance Mtb infection and disease progression as both infections drive contrasting immune responses in the host. Helminth infection skews the host immune system to a Th2 response and impairs Th1 response needed to overcome Mb infection [66]. Anti-helminthic treatment in patients with latent TB has been shown to reverse this condition with a restored Th1 response [67], supporting the influence of helminth co-infection in TB patients. In another study, anti-helminthic-treated asymptomatic helminth carriers showed superior immunogenicity to BCG vaccination compared to untreated carriers [68]. This suggests that intestinal infection could also influence the efficacy of TB vaccines. The modulation of macrophage immunometabolism upon helminth infection has been comprehensively reviewed recently, with similar findings of M2 macrophage polarization that shift their metabolism predominantly to OXPHOS, lipid oxidation, and amino acid metabolism [69], which may also influence the immune response in the lung during co-infection with Mtb.

Growing evidence of the influence of gut microbiota in the outcome of Mtb infection further highlights the importance of the gut-lung axis to protect against TB. In a mouse model, dysbiosis of gut microbiota by wide-spectrum antibiotics has been shown to enhance Mtb colonization in the lung [70]. Reduction of MAIT cell populations with reduced production of IL-17A was observed in the lungs of antibiotic-treated mice after one week of Mtb infection, which likely contributed to the enhanced Mtb colonization [70]. In other words, a gut microbiota-dependent depletion of MAIT cells may weaken the host’s early immune response to Mtb infection, suggesting a gut-lung axis that may influence the outcome of Mtb growth.

It would be interesting to explore if mucosal vaccination against Mtb would provide a better outcome of protection, not only because of the delivery through the portal of entry of Mtb *per se* but the ability to harness the gut’s innate and adaptive immunity to directly influence immunity in the lungs. In line with this idea, our research group is exploring an oral live attenuated *Vibrio cholerae* as a candidate vaccine against *V. cholerae* with the ability to be a delivery vector for a DNA vaccine carrying heterologous TB antigens.

*Vibrio cholerae* can colonize the surface of the epithelial cells of the small intestine, despite competition with normal microbiota, due to its ability to adjust its genes to respond to stress in terms of Type VI secretion system, quorum sensing, reactive oxygen species/pH, and bioactive metabolites [71]. Leung et al. reported that in the acute phase of cholera, circulating MAIT cells were activated, reflecting their involvement in the innate immune response to cholera. The proportion of activated MAIT cells was also found to be increased as the disease progressed, which was positively correlated with an increase in anti-LPS IgA and IgG, but not IgM, suggesting a role played by MAIT cells in LPS antibody responses, potentially with a specific role in antibody class switching [72]. The induction of Mtb-specific immune cells, including MAIT cells in the gut, may have exciting implications for the control of TB that deserve to be explored.

## 6. Summary

The interaction between Mtb and humans has persisted for more than 70,000 years [73], which reflects the complex interplay between them. With recent advancements in technology, our ability to fill the knowledge gaps has been greatly enhanced. We have started to appreciate that the host immune response to Mtb infection in the lung could be influenced by immune cells’ metabolic activity as observed during the acute and chronic phases of TB infection. Anti-inflammatory immune cells and cells in the basal state are predominantly maintained by OXPHOS. Once the cells are activated and differentiated into a pro-inflammatory phenotype upon infection by Mtb, their metabolic activity is switched to aerobic glycolysis that produces energy more rapidly, albeit less efficiently, and generates components for their effector functions. Yet Mtb can remain dormant within the host with the potential of reactivation when the opportunity arises.

The complex nature of immunometabolism communications at the site of infection as discussed above suggests that mucosal delivery of future TB therapeutics and vaccines should be explored more extensively. Growing evidence of cross-talk between lung and gastrointestinal immunity strengthens the idea of developing an oral vaccine candidate for TB, which is a widely preferable route of vaccine delivery for mass vaccination.

## Figures and Tables

**Figure 1 ijms-23-08531-f001:**
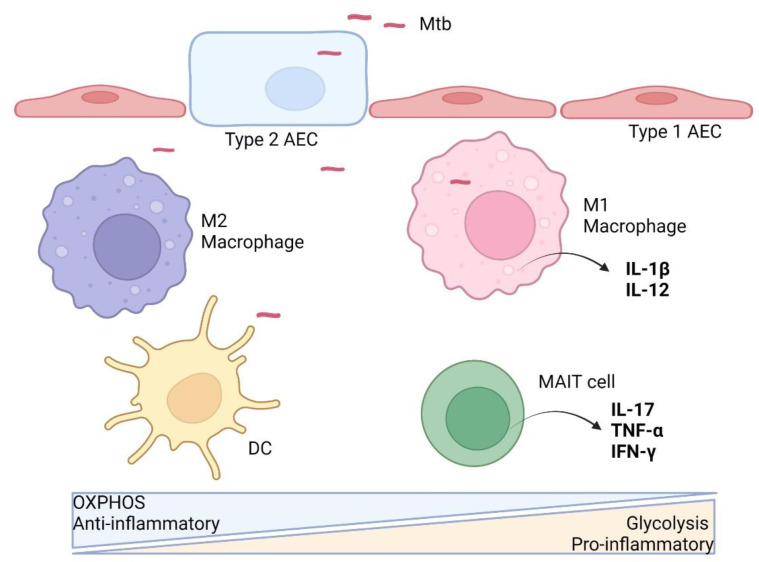
Innate immune cell immunometabolism against Mtb. After aerosol infection and successful breach of the alveolar epithelial cells (AEC), Mtb is taken up by resident alveolar macrophages (M2) and interstitial dendritic cells (DC), which predominantly rely on oxidative phosphorylation (OXPHOS) metabolism at the basal state without anti-microbial activities. Upon stimulation, the macrophage is activated, switching to aerobic glycolysis metabolism and a pro-inflammatory M1 phenotype, with the release of cytokines (e.g., IL-12) to activate more immune cells. Resident MAIT cells may also get activated upon Mtb infection with enhanced glycolytic metabolism and anti-microbial phenotype. Activated DCs migrate to draining lymph nodes to activate T cells. (Illustration created with BioRender.com, accessed on 19 June 2022).

**Figure 2 ijms-23-08531-f002:**
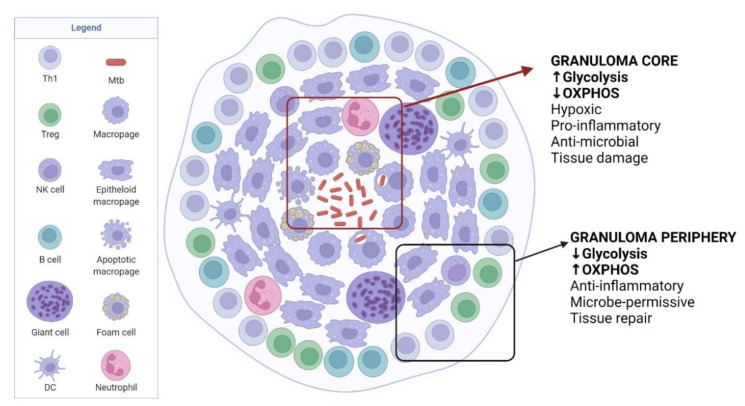
TB immunometabolism in the granuloma. Granuloma develops as TB disease progresses, which contains Mtb but also helps Mtb to persist in the host. The core of granuloma is highly hypoxic and inflammatory, mainly maintained by glycolysis, to kill Mtb but causes damage to host tissues. Moving to the granuloma periphery, the environment becomes less inflammatory with increased tissue repair functions but favourable to Mtb survival. Adapted from “Granuloma”, by BioRender.com (2022). Retrieved from https://app.biorender.com/biorender-templates, accessed on 19 June 2022.

## Data Availability

Not applicable.

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
