# Peer review of "Immunometabolism of Immune Cells in Mucosal Environment Drives Effector Responses against Mycobacterium tuberculosis"

_ijms, 2022, doi:10.3390/ijms23158531_

Round 1

Reviewer 1 Report

Tukami and Norazmi in their manuscript entitled "Immunometabolism of immune cells in mucosal environment drive effector responses against Mycobacterium tuberculosis" present a timely overview of the immunometabolism involved in Mtb infection. Overall I find the review to be well written and would accept acceptance in the current form with minor revisions to spell check and grammar.

Author Response

Thanks for your kind comments. We have made the necessary revisions, to check the spelling and grammar as suggested.

Reviewer 2 Report

1. The description of mucosal environment 2 drive effector responses from the title is very lacking.

Please edit the title or add specifics to 2. Immunometabolism During Early Phase of Mtb Infection section and 3. Immunometabolism During Chronic Phase of TB section, should be described in detail, to increase reader’s understanding.

2. Please add recent papers and key papers in related fields.

Author Response

We appreciate the reviewer’s comments on the lack of information on the specific topics. Actually, the literature is scarce with respect to the topics being highlighted. Despite this, we have included a fairly detailed review of the different immune cells that are probably involved in the early and chronic phases of the infection. Due to the scarcity of the literature, further deliberation on the topic would render it overly speculative. As such, we are not able to cite additional recent or key papers on the specific points as suggested. In fact, we have already included references to a few recent reviews to help the readers gain more detailed insights into the general understanding on the topic. The positive comments on the novelty of our review by the other reviewers attest to the scarcity of the literature on this exciting area.

Reviewer 3 Report

In this interesting review the authors summarize the current knowledge on immunometabolism of immune cells in mucosal environment 2 drive effector responses against Mycobacterium tuberculosis.  As the authors pointed out a better understanding on mucosal immunometabolism of resident effector cells in response to Mycobacterium tuberculosis infection and how the pathogen manipulates them for its survival and growth, which could address our knowledge gaps while opening up new questions, and potentially be applied for future vaccination and therapeutic strategies. They also address the (scarce) indirect evidences of the possible impact of the gut-lung axis on protection against Mycobacterium tuberculosis.

The subject is novel and there are only one specific reviews on it, cited by the authors, and with a more basic and less applied approach (Kumar R, et al. Front Mol Biosci. , 2019). The subject is novel and there are no specific reviews on it. The most relevant evidence on the research question is included and the design of the review and supporting material (figures) are appropriate.

There are only minor problems that could be corrected to improve the understanding of the manuscript, which I point out below.

The authors should describe all abbreviations used the first time they appear. For instance: MAIT (line 54), DC line 60), AEC (line 58), etc.

AEC is airway epithelial cells (line 70) or Alveolar Epithelial Cells (line 68). Plese, be consistent.

Given the large number of abbreviations used, I suggest including a glossary of abbreviations at the beginning of the manuscript.

Author Response

Thanks for pointing out the difference in phrases used for AEC. We have made the correction as needed. We have also added a list of abbreviations as suggested, at the end of the manuscript.

Reviewer 4 Report

I would like to congratulate the authors of the article for the good and clear exposition of the topic.

The only thing I would dare to suggest, but I don't know if it is within the editorial policy of the journal, is that since many acronyms are used, it would be good if there were an index of them, so it would make reading much easier, especially for those who are neophytes in tuberculosis immunity.

Author Response

Thanks for your kind comments. We have added a list of abbreviations as suggested, at the end of the manuscript.

Round 2

Reviewer 2 Report

My concerns has been addressed.